



# Miocene basement exhumation in the Central Alps recorded
# by detrital garnet geochemistry in foreland basin deposits
**Laura Stutenbecker[1]\*, Peter M.E. Tollan[2], Andrea Madella[3], Pierre Lanari[2]**
*[1]Institute of Applied Geosciences, Technische Universität Darmstadt, Schnittspahnstr. 9,*
*64287 Darmstadt, Germany*
*[2]Institute of Geological Sciences, University of Bern, Baltzerstrasse 1+3, 3012 Bern,*
*Switzerland*
*[3]Department of Geosciences, University of Tuebingen, Wilhelmstr. 56, 72074 Tübingen,*
*Germany*
\*corresponding author: stutenbecker@geo.tu-darmstadt.de
Abstract
The Neogene evolution of the European Alps was characterized by the exhumation of crystalline
basement, the so-called external crystalline massifs. Their exhumation presumably controlled the
evolution of relief, distribution of drainage networks and generation of sediment in the Central Alps.
However, due to the absence of suitable proxies, the timing of their surficial exposure, and thus the
initiation of sediment supply from these areas, are poorly constrained.
The northern alpine foreland basin preserves the Oligocene to Miocene sedimentary record of tectonic
and climatic adjustments in the hinterland. This contribution analyses the provenance of 25 to 14 My-
old alluvial fan deposits by means of detrital garnet chemistry. Unusually grossular- and spessartine-
rich garnets are found to be unique proxies for identifying detritus from the external crystalline
massifs. In the foreland basin, these garnets are abundant in 14 My-old deposits, thus providing a
minimum age for the surficial exposure of the crystalline basement.



## 1. Introduction

Tectonic processes influence the evolution of relief in mountain chains and consequently control the development of the drainage network, sediment supply and deposition in the foreland basin. The Central European Alps and their northern foreland basin, formed through the collision of the European and the Adriatic continents since the Eocene (Schmid et al. 1996, Handy et al. 2010), are a classic example of such interactions (e.g. Schlunegger et al., 1998; Pfiffner et al., 2002; Vernon et al., 2008, 2009; Baran et al., 2014; Fox et al., 2015). The exhumation of large slices of mid-crustal rocks from the European plate, the so-called external crystalline massifs, occurred during a late-stage orogenic event, possibly controlled by crustal delamination in response to lithospheric mantle rollback (Herwegh et al., 2017). The areas exhumed during this event are today characterized by high relief, intense glaciation and some of the highest denudation rates measured in the Alps, which all contribute to their importance as a sediment source (Kühni and Pfiffner, 2001; Wittmann et al., 2007; Stutenbecker et al., 2018).

Peak metamorphism of lower to upper greenschist-facies conditions occurred between 17 and 22 Ma in all external crystalline massifs (Mont Blanc, Aar massifs and the Gotthard nappe, Challandes et al., 2008; Rolland et al., 2008; Cenki-Tok et al., 2014; Nibourel et al., 2018). Their subsequent exhumation has been investigated using thermochronology by a number of studies (e.g. Schaer et al. 1975, Wagner et al. 1977; Michalski and Soom, 1990; Vernon et al., 2009; Glotzbach et al., 2010). While some studies concluded that exhumation was episodic (Vernon et al. 2009), others suggest relatively constant exhumation rates of 0.5-0.7 km/My since 14 My (Michalski and Soom, 1990; Glotzbach et al., 2010). The timing of the first surficial exposure of the external massifs has, however, never been constrained, because estimates of their total thickness have not yet been established. In most geometric reconstructions (e.g. Pfiffner, 1986, 2017; Schmid et al., 2004), the contact between the crystalline basement and the overlying Mesozoic cover is assumed to be relatively flat, and the top of the crystalline basement is hypothesized to be less than one kilometer above the modern topography. Conversely, a new reconstruction of this tectonic contact allows for a substantially greater amount (~8 km) of (now eroded) crystalline rock on top of the present-day topography (Nibourel et al., 2018).

This study aims to constrain the timing of exposure, and thus the beginning of sediment supply from the external crystalline massifs, by determining the provenance of the foreland basin deposits. Sediments preserved in the northern peripheral foreland basin of the Central Alps, the Swiss part of the Molasse basin, are a well-studied archive recording tectonic and climatic adjustments in the central orogen between ca. 32 and 14 My ago (Schlunegger et al., 1993, 1996; Kempf et al., 1999; Spiegel et al., 2000; Kuhlemann and Kempf, 2002; von Eynatten, 2003; Schlunegger and Kissling, 2015). So far, the provenance of the Molasse deposits has been investigated using optical heavy mineral analysis, framework petrography and both bulk and single-grain geochemical techniques, including epidote geochemistry and cooling ages derived from zircon fission track analysis and Ar/Ar dating of white mica (Spiegel et al., 2000, 2002; von Eynatten, 2003; von Eynatten and Wijbrans, 2003). Conclusive evidence for a contribution from the external crystalline massifs, however, has remained elusive, leading to the assumption that their exposure must post-date the youngest preserved (ca. 14 My-old) Molasse sediments (von Eynatten, 2003).

In this study, we use detrital garnet major element geochemistry in Miocene deposits preserved in the central part of the Swiss foreland basin. The great compositional variability displayed by garnet from different source rocks means that it is a useful provenance tracer in a variety of settings (Spear, 1994; Mange and Morton, 2007). Furthermore, it is a common heavy mineral in (orogenic) sediments (Garzanti and Andò, 2007) and is relatively stable during transport and diagenesis (Morton and



Hallsworth, 2007). In the Central Alps, detrital garnet has recently been shown to be a valuable provenance indicator, especially for distinguishing detritus supplied from the external crystalline massifs (Stutenbecker et al., 2017). We aim (1) to provide additional provenance information to unravel the Miocene history of the Molasse deposits and its tectonic forcing and (2) to test whether detritus from the external massifs is present in the younger Molasse deposits in order to give independent constraints on the timing of crystalline basement exhumation.

## 1.1 Geological Setting

The Central Alps evolved through convergence between the European continental margin in the north and the Adriatic plate in the south (Schmid et al., 1996). Convergence started in the late Cretaceous with the subduction of the alpine Tethys ocean below the Adriatic microplate (Froitzheim et al., 1996), and ceased in the Paleogene after the European continental lithosphere entered the subduction zone. These Cretaceous to early Neogene orogenic processes are reflected by the syn-orogenic deposition of deep-marine flysch units preserved throughout the Alps (see e.g. Wildi, 1985; Winkler 1996). Around 32 Ma ago, the sedimentation style in the northern foreland basin changed from marine, flysch-like deposition to shallow marine and terrestrial sedimentation. This is thought to represent the transition to Molasse-type sedimentation in an overfilled basin and is discussed to be potentislly related to a breakoff of the European slab around the time of the Eocene-Oligocene boundary (e.g. Sinclair et al. 1991; Sinclair 1997; Schlunegger and Kissling, 2015). Since this time, the northern foreland basin has become a major sink of orogenic detritus and an important sedimentary archive.

The sediments in the Swiss part of the northern foreland basin are divided into four litho-stratigraphic units that represent two shallowing- and coarsening-up megacycles (Schlunegger et al., 1998). The first cycle consists of the Rupelian Lower Marine Molasse (LMM) and the Chattian and Aquitanian Lower Freshwater Molasse (LFM). The second megacycle comprises a transgressive facies of Burdigalian age (the Upper Marine Molasse, UMM) overlain by Langhian to Serravalian deposits of the Upper Freshwater Molasse (UFM). The depositional ages of these units were constrained using mammal biostratigraphy and magnetostratigraphy (Engesser, 1990; Schlunegger et al., 1996). Throughout the Oligocene and the Miocene, the proximal Molasse deposits are thought to have been formed through a series of large alluvial fans (Fig. 1) aligned along the alpine thrust front (Schlunegger et al., 1993; Kuhlemann and Kempf, 2002). The more distal parts of the basin were instead characterized by axial drainage directed towards the Paratethys in the East/Northeast (31-20 My) and the Western Mediterranean Sea in the Southwest (after 20 My), respectively (Kuhlemann and Kempf, 2002). Whereas the more distal deposits could be significantly influenced by long-distance transport from the northeast or southwest, the alluvial fans are thought to carry a local provenance signal from the rocks exposed immediately south of each fan system due to their proximal nature.

The hinterland of the central Swiss foreland basin comprises, from north to south, potential source rocks derived from the following architectural elements (Figs. 1, 2):

(1) The Prealps Romandes; a stack of non-metamorphic and weakly metamorphosed sedimentary cover nappes (Mesozoic carbonates and Cretaceous-Eocene flysch), interpreted as the accretionary wedge of the alpine Tethys, detached from its basement and thrusted northwards onto the European units.

(2) The Helvetic nappes; the non- or very low-grade metamorphic sedimentary cover sequence of the European continental margin (mostly Mesozoic carbonates).

(3) The external crystalline massifs; lentoid-shaped autochthonous bodies of European continental crust that consist of a pre-Variscan polycyclic gneiss basement intruded by Upper Carboniferous to Permian granitoid rocks and an overlying metasedimentary cover. They were buried within the Alpine nappe stack in the Oligocene (Cenki-Tok et al., 2014), reaching





greenschist facies peak-metamorphic conditions between 17 and 22 My ago (Fig. 2) and were
exhumed during the Miocene. The Gotthard nappe, although not a "massif" *sensu stricto*
because of its allochthonous nature, will be included into the term "external crystalline
massifs" from here on, because the timing and the rates of exhumation are comparable
(Glotzbach et al., 2010).
(4) The Lepontine dome; an allochthonous nappe stack of European Paleozoic gneiss basement
and its Mesozoic metasedimentary cover (Berger et al. 2005). Amphibolite-facies peak
metamorphism (Frey and Ferreiro Mählmann 1999, Fig. 2) in the Lepontine occurred
diachronously at around 30-27 My ago in the south (Gebauer, 1999) and as late as 19 My ago
in the north (Janots et al., 2009). Although the onset of exhumation of the Lepontine dome
might have been equally diachronous, it is generally assumed to have occurred before 23 My
ago (Hurford, 1986).
(5) The Penninic nappes, containing ophiolites of the alpine Tethys as well as the continental crust
of Briançonnais, a microcontinent located within the alpine Tethys between the southern
Piedmont-Ligurian ocean and the northern Valais trough (Schmid et al. 2004).
(6) The Austroalpine nappes, containing the basement and sedimentary cover of the Adriatic plate
with a Cretaceous ("Eoalpine", ca. 90-110 My) metamorphic peak of greenschist facies
conditions (Schmid et al. 2004). Although the Austroalpine nappes are found exclusively in
the Eastern Alps to the east of the Lepontine dome today, we mention them here as well,
because they were probably part of the nappe stack in the Central Alps prior to their erosion
during the Oligocene and Miocene.
(7) The Sesia/Dent Blanche nappe, probably representing rifted segments of the basement and
sedimentary cover of a distal part of the Adriatic plate (Froitzheim et al. 1996). In contrast to
the Austroalpine nappes, the Sesia/Dent Blanche nappe was subducted and exposed to
blueschist-facies (Bousquet et al. 2012, Fig. 2) to eclogite-facies metamorphism (e.g.
Oberhänsli et al. 2004).

### 1.2 Compositional trends in the Napf fan

Rocks from the Central Alps are generally considered as the major sediment source of all proximal
Molasse basin deposits, while compositional changes in the foreland are thought to directly reflect
tectonic and erosional processes in the immediate alpine hinterland (Matter, 1964; Schlunegger et al.,
1993; 1998). The compositional evolution in the basin is diachronous and not uniform between the
different fan systems (e.g. Schlunegger et al., 1998; Spiegel et al., 2000; von Eynatten, 2003). In this
study, we will focus on the Napf fan, located in the central part of the basin, which is the most likely to
archive a provenance signal related to external massif exhumation due to its proximity to the large
crystalline basement slices of the Aar massif and the Gotthard nappe (Fig. 1). In the Napf fan, three
major compositional trends have been previously identified (Fig. 3):
(Phase 1) Between ~31 and ~25 My ago, the heavy minerals are dominated by the zircon-tourmaline-
rutile (ZTR) assemblage and garnet (von Eynatten, 2003). Rock fragments are dominantly of
sedimentary origin and zircon fission track ages are Paleozoic to late Mesozoic (Spiegel et al., 2000).
This phase is consistently interpreted by different authors to reflect the erosion of (Austroalpine)
flysch-like sedimentary cover nappes, which are structurally highest in the central alpine nappe stack
(Schlunegger et al., 1998; Spiegel et al., 2000; von Eynatten, 2003).
(Phase 2) 25-21 My ago: Around 25 My ago, the occurrence of epidote as well as an increase in
granitic rock fragments mark a major compositional change in the foreland. The presence of
characteristic colorful granite pebbles suggests an origin from the Austroalpine Bernina nappe (Matter,





1964). Sediments of this phase clearly reflect the down-cutting into crystalline basement and are
consistent with a continuation of a normal unroofing sequence. Additionally, (Schlunegger et al.,
1998) report the occurrence of quartzite pebbles, possibly sourced from the Penninic Siviez-Mischabel
nappe and argue that parts of the epidote could originate from Penninic ophiolites as well, thus
suggesting that erosion might have already reached down into the Penninic nappes. Spiegel et al.,
(2002) argued against this Penninic contribution based on the $^{87}Sr/^{86}Sr$ and $^{143}Nd/^{144}Nd$ isotopic
signatures of the epidote.
(Phase 3) 21-14 My ago: At ~21 My, metamorphic rock fragments occur in the sediments, while the
heavy mineral assemblages remain epidote-dominated and overall similar to the second phase. Zircon
fission track ages are exclusively Cenozoic (ages peaks between ~32 and ~19 Ma). In contrast to the
first two phases, the sediment composition allows several, partially contradicting interpretations.
Whilst petrographical and mineralogical data might suggest recycling and sediment mixing (von
Eynatten, 2003), young $^{40}Ar/^{39}Ar$ cooling ages in white mica (von Eynatten, 2003; von Eynatten and
Wijbrans, 2003) and exclusively young zircon fission track ages (Spiegel et al., 2000) point to an
additional, newly exhumed source that these authors identify as the Lepontine dome. Based on the
abundance of flysch pebbles after ~21 My, Schlunegger et al. (1998) favor an alternative scenario, in
which the erosional front shifted northwards into the flysch nappes of the Prealps Romandes.
Furthermore, the isotopic signature of detrital epidotes suggests a contribution of mantle source rocks
between ca. 21 and 19 My ago, which could point to a contribution by Penninic ophiolites (Spiegel et
al., 2002). However, this is not reflected in the heavy mineral spectra (von Eynatten, 2003), which do
not contain typical ophiolite minerals such as Cr-spinel.
In none of these scenarios were the external crystalline massifs considered as a possible sediment
source. The exact time of their surficial exposure is unknown, but it is believed to post-date the
youngest preserved Molasse sediments. This interpretation is based on the lack of granitic pebbles
attributable to the external massifs in the Molasse (Trümpy, 1980) and on structural reconstructions
(e.g. Pfiffner, 1986) in combination with thermochronological data (e.g. Michalski and Soom, 1990).

## 2. Sampling strategy and methodology

In order to characterize the detrital garnets in the foreland, three samples were taken from 25 My-, 19
My- and 14 My-old fine- to medium-grained fluvial sandstones within the Napf fan deposits located
ca. 40 kilometers to the East and Southeast of Bern in the central part of the Swiss Molasse basin. The
exact sampling sites were chosen based on the availability of published petrographical, chemical and
mineralogical data (von Eynatten, 2003) as well as magnetostratigraphic calibration (Schlunegger et
al., 1996).
Because the potential source rocks were already narrowed down to particular regions based on other
provenance proxies, and because many of these rocks are still preserved in the alpine chain today, it is
possible to compare potential source compositions to the detrital ones. For comparison we used
detrital data from Stutenbecker et al. (2017) as well as published source rock data from different units
across the Central Alps (Steck and Burri, 1971; Chinner and Dixon, 1973; Ernst and Dal Piaz, 1978;
Hunziker and Zingg, 1980; Oberhänsli, 1980; Sartori, 1990; Thélin et al., 1990; Reinecke, 1998; von
Raumer et al., 1999; Cartwright and Barnicoat, 2002; Bucher and Bousquet, 2007; Angiboust et al.,
2009; Bucher and Grapes, 2009; Weber and Bucher, 2015).
In addition, three river sand samples were collected from small monolithological catchments (3-30
km$^2$) draining garnet-bearing potential source rocks that were previously not, or only partially,
considered in the literature. We prefer this "tributary sampling approach" (see e.g. Stutenbecker et al.,
2017) over sampling specific source rocks, because small monolithological catchments are more likely





to comprise all garnet varieties of the targeted source rock and to average out differences in garnet
fertility. The targeted source areas are located within the Gurnigel flysch (Prealpes Romandes), the
Antigorio nappe orthogneisses of the Lepontine dome, and the Lebendun nappe paragneisses of the
Lepontine dome (Fig.1). Sample characteristics are summarized in Table 1 and Table 2. For detailed
lithological descriptions of the sampled sites in the Napf area, see Schlunegger et al. (1993) and von
Eynatten (2003).
The sandstone samples were carefully disintegrated using a jaw breaker and a pestle and mortar. The
disintegrated sandstones as well as the source rock tributary sands were sieved into four grain size
classes of <63 μm, 63-125 μm, 125-250 μm and >250 μm. The fractions of 63-125 μm and 125-250
μm were further processed in sodium polytungstate heavy liquid at 2.85 g/cm$^3$ to concentrate heavy
minerals. The heavy mineral concentrates were dried and, depending on the obtained amounts, split
into 2-4 parts using a microsplitter. All measured garnet grains were hand-picked from the concentrate
of one split part per fraction under a binocular microscope.
The grains were subsequently arranged in lines on sticky tape, embedded into epoxy resin, ground
with SiC abrasive paper (grits 400, 800, 1200, 2500, 4000), polished using 3, 1 and ¼ μm diamond
suspensions and graphite-coated. Major element oxides were analyzed using a JEOL JXA-8200
electron probe micro-analyzer at the Institute of Geological Science at University of Bern,
Switzerland, under standard operating conditions for garnet (see Giuntoli et al., 2018): accelerating
voltage of 15 KeV, electron beam current of 15 nA, beam diameter of 1μm, 20 s peak acquisition time
for Si, Ti, Al, Fe, Mn, Mg, Ca and 10 s for both backgrounds. Natural and synthetic standard olivine
(SiO$_2$, MgO, FeO), anorthite (Al$_2$O$_3$, CaO) ilmenite (TiO$_2$) and tephroite (MnO) were used for
calibration by applying a CITIZAF correction (Armstrong, 1984). Garnet compositions were measured
as close as possible to the geometric centers of grains, unless the area was heavily fractured. In some
randomly selected grains core and rim compositions were measured to identify intra-grain chemical
variability; these core/rim pairs are reported separately in Stutenbecker (2019).
Molecular proportions were calculated from the measured main oxide compositions on the base of 12
anhydrous oxygens. Because ferric and ferrous iron were not measured separately (FeO = Fe$_{total}$), the
Fe$^{2+}$/Fe$^{3+}$ ratio was determined based on charge balance (Locock, 2008). Garnet endmember
compositions were subsequently calculated using the Excel spreadsheet by Locock (2008). Garnet is a
solid solution between different endmembers, the most common ones being almandine (Fe$_3$Al$_2$Si$_3$O$_{12}$),
grossular (Ca$_3$Al$_2$Si$_3$O$_{12}$), pyrope (Mg$_3$Al$_2$Si$_3$O$_{12}$), spessartine (Mn$_3$Al$_2$Si$_3$O$_{12}$) and andradite
(Ca$_3$Fe$_2$Si$_3$O$_{12}$). The relative proportions of these endmember components depend on bulk rock
composition and intensive parameters (such as temperature and pressure) which can vary substantially
depending on the metamorphic or magmatic history of the protolith (Deer et al., 1992; Spear, 1994).
The data were plotted and classified using the ternary diagram of Mange and Morton (2007) as well as
the linear discriminant function method of Tolosana-Delgado et al. (2018) based on a global data
compilation on garnet compositions from different source rocks (Krippner et al., 2014).
3. Results
Most of the detrital garnets are dominated by the Fe-rich almandine endmember with varying amounts
of grossular, pyrope, spessartine and andradite (Fig. 4). Other endmembers (e.g. uvarovite) are
negligible. Minimum, maximum and average endmember contents are summarized in Table 3; for the
full dataset we refer to Stutenbecker (2019). Garnet compositions do not differ significantly between
the two analyzed grain size fractions of the same sample, although some slight variations are visible in
the ternary plot (Fig. 4): In sample LS2016-18 (25 My, Fig. 4a) garnets of the 125-250 μm fraction
tend to be enriched in pyrope with respect to garnets of the 63-125 μm fraction. In sample LS2018-5





(19 My, Fig. 4b) 4 "outliers" that are very pyrope- and grossular-rich (n=2) or grossular- and
andradite-rich (n=2) occur only in the 63-125 μm grain size fraction. Furthermore, garnet grains of the
63-125 μm fraction are more frequently grossular-rich compared to the 125-250 μm fraction. In
sample LS2017-3 (14 My, Fig. 4c), the 63-125 μm fraction contains some garnet grains (n=8) of high
almandine and low grossular content that are absent in the 125-250 μm fraction.
Although some individual garnets show distinct internal compositional zoning from core to rim, the
intra-grain chemical variability is generally negligible (see Stutenbecker, 2019).
According to the ternary classification plot of Mange and Morton (2007), the major part of garnet in
all three samples (>80 %) belong to the B-type and thus point to a dominant amphibolite-facies source
rock (Table 4). Minor portions are derived from C-type (high-grade metabasic), A-type (granulite
facies) and D-type (metasomatic) sources. Classification through linear discriminant analysis
(Tolosana-Delgado et al., 2018) yields a similar trend with generally high proportions of amphibolite-
facies source rocks (class B-garnets, >70 %, Table 4). Some grains (5 %, 3 % and 12 % in the 25 My-,
19 My- and 14 My-old samples, respectively) were classified as igneous garnet (Table 4).
Distinct compositional changes between the 25 My-, 19 My- and 14 My-old Molasse sediments are
mostly related to the ratio of almandine and grossular contents (Table 3, Fig. 5). At 25 My, garnets are
dominantly almandine-rich (average 70 %) and grossular-poor (average 9 %). At 19 My, both
grossular-poor and grossular-richer garnets occur (average 16 %). Garnets in the 14 My-old sample are
generally almandine-poorer (average 50 %) and grossular-rich (average 32 %). This implies (1) that
garnets contained in the younger sediment (14 and, to some extent, 19 My) were not recycled in
significant amounts from the older Molasse strata and (2) that at least two sources supplied B-type
garnets during Molasse deposition.
Garnet compositions from the three potential source rock samples analyzed in this study are shown in
Fig. 4d (Lepontine paragneiss and Lepontine orthogneiss) and Fig. 4e (Gurnigel flysch). The average
compositions are displayed in Fig. 6; for the full dataset we refer to Stutenbecker (2019). Likewise,
average compositions of garnet from the literature (external massif granite garnets, eclogite facies
garnets and granulite facies garnets) are displayed in Fig. 6.
All source rocks, except for the external crystalline massif granites, supply almandine-dominated (i.e.
>50 % almandine-component) garnet. The andradite content in all source rock garnets is very low, but
they contain varying amounts of grossular, spessartine and pyrope. Garnets from the Lepontine
gneisses (Fig. 4d) are generally almandine-rich, but those in the paragneiss tend to be grossular-richer
(22 %) compared to the ones in the orthogneiss (11 %). The Gurnigel flysch garnets (Fig. 4e) are
almandine-rich with elevated pyrope contents (14 %). Garnets from the external crystalline massifs
(Fig. 4f) are unusually rich in grossular (35 %) and spessartine (21 %), and the almandine content is
much lower than in the other source rock garnets (34 %). Eclogite-facies garnets have high grossular
(23 %) and pyrope (16 %) contents (Fig. 4g). Granulite-facies garnets (Fig. 4g) have on average the
highest pyrope content of all source rock garnets (25 %).
4. Discussion
**4.1 Origin of amphibolite-facies garnets**
According to the compositional classification of Mange and Morton (2007) and Tolosana-Delgado et
al. (2018), the majority of detrital garnet grains in the Molasse were derived from amphibolite-facies



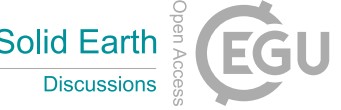

source rocks ("B-type"). In the Central Alps, amphibolite-facies conditions of alpine age were only
reached in the Lepontine nappes (Fig. 2). However, many gneisses in the area preserve a pre-Mesozoic
amphibolite-facies metamorphic signature as well (Frey et al., 1999), for example in the Austroalpine
Bernina nappe (Spillmann, 1993; Spillmann and Büchi, 1993), the middle Penninic Briançonnais
basement (Sartori et al., 2006) or the polycyclic basement of the external massifs (von Raumer et al.,
1999). In fact, the Gurnigel flysch, a Late Cretaceous to Eocene flysch nappe in the Prealps Romandes
that did not undergo alpine metamorphism (Fig. 2), contains almost exclusively almandine-rich B-type
garnets (Fig. 4e).
These considerations indicate that, following the classification scheme of Mange and Morton (2007)
alone, the provenance of Alpine B-type garnets remains ambiguous. However, petrographic findings
as well as zircon fission-track analysis and Ar/Ar dating in white mica (Spiegel et al., 2000; von
Eynatten, 2003; von Eynatten and Wijbrans, 2003) strongly suggest a compositional change ca. 21 My
ago towards a metasedimentary source with a young cooling history. These authors relate this shift to
the erosion of the sedimentary cover of the Lepontine dome. Source rock samples taken within the
Lepontine dome from the crystalline basement (Antigorio nappe orthogneiss) and the meta-
sedimentary cover (Lebendun nappe paragneiss) contain generally almandine-rich garnets, but those
from the paragneiss tend to be richer in grossular than those from the orthogneisses (Fig. 4). Because
the amount of grossular-rich garnet is higher in the 19 My-old sample compared to the 25 My-old
sample, the data could support an origin from the Lepontine meta-sedimentary cover.

### 4.2 Origin of granulite-facies garnets

Granulite-facies garnet grains with relatively high pyrope and low grossular contents ("A-type" and
"Class C" garnets according to Mange and Morton (2007) and Tolosana-Delgado et al. (2018),
respectively) are only frequent in the 19 My-old Molasse sample (ca. 8-9 %, Table 3). Alpine
Granulite-facies metamorphic conditions in the Central Alps were only reached in the Gruf complex
located close to the Insubric line between the Lepontine dome and the Bergell intrusion (Fig. 2).
Furthermore, there is evidence for pre-Mesozoic granulite-facies metamorphism in some rocks in the
Ivrea zone south of the Insubric line (Hunziker and Zingg, 1980), in the Sesia Zone (Engi et al., 2018;
Giuntoli et al., 2018) and in the Dent Blanche nappe (Angiboust et al., 2009). It is unlikely that erosion
reached so far to the South during the Miocene, because the Penninic and probably also the exhuming
Lepontine nappe stack would have acted as a topographic barrier to the fluvial drainage network.
However, it was proposed that the flysch sediments preserved in the Prealps Romandes were partially
fed by these units during the Late Cretaceous and the Eocene (Wildi, 1985; Ragusa et al., 2017). This
interpretation is supported by the Gurnigel flysch sample (Fig. 4e), which contains garnet of granulite-
facies type. A recycled flysch origin is supported further by the abundance of flysch sandstone pebbles
in Molasse strata of the same age (Schlunegger et al., 1998).

### 4.3 Origin of eclogite-facies garnets

The 19 My-old sample contains two grains with very high pyrope contents classified as eclogite-facies
garnets ("Ci-type" and "Class A" garnets according to Mange and Morton (2007) and Tolosana-
Delgado et al. (2018), respectively). In the 14 My-old sample, 15 % of grains were classified as
eclogite-facies garnets as well following the Mange and Morton (2007) approach, although these
grains are pyrope-poorer compared to the garnets in the 19 My-old sample and therefore probably
originate from a different source. Eclogite-facies garnets are known from metamorphic rocks of the
upper Penninic alpine ophiolites (e.g. Bucher and Grapes, 2009; Weber and Bucher, 2015, Fig. 2), but
also from Paleozoic (?) gneisses of the middle Penninic Briançonnais basement (Sartori, 1990; Thélin
et al., 1990). In addition, the Gurnigel flysch contains a minor, pyrope-rich population of garnets (Fig.
4e). Detrital garnet compositions similar to the ones in the 14 My-old sample were reported from the
Goneri river draining the Gotthard nappe (Stutenbecker et al., 2017).





Overall the eclogitic component in all three samples is only minor, thus supporting results from von
Eynatten (2003), who concluded that eclogites were not a major source during the Miocene in the
Napf area.

### 4.4 Origin of "igneous" garnets

Of the garnets from the youngest, 14 My-old Molasse sample, 12 % can be classified as igneous
("Class E", Table 4) according to Tolosana-Delgado et al. (2018). Their high grossular and very low
pyrope content distinguishes them clearly from all the other, generally more almandine-rich, garnets.
In the classification scheme after Mange and Morton (2007), however, this type of garnet plots in the
D-type or in the rightmost part of the B-type or field (Fig. 4, Table 4). The detrital garnet signature of
the 14 My-old sample mirrors almost exactly the compositional range of garnets from the external
crystalline massifs (Fig. 4c, 4f). In the external crystalline massifs, these garnets grew in Permo-
Carboniferous plutons under alpine greenschist-facies metamorphic conditions (Steck and Burri, 1971,
Fig. 2). They are restricted to the granitoid basement of the external massifs and do not occur
anywhere else in the Central Alps, which makes them an excellent provenance proxy (Stutenbecker et
al., 2017). A further distinction among garnets supplied by the different plutons (e.g. the Central Aar
granite from the Aar massif, the Rotondo granite from the Gotthard nappe or the Mont Blanc granite
from the Mont Blanc massif) is not possible based on garnet major element geochemistry alone
(Stutenbecker et al., 2017).

### 4.5 Implications for the evolution of the Alpine orogen

Previous provenance studies have identified meta-sedimentary detritus in the youngest (ca. 14 My old)
Molasse and located its source in the unroofing sedimentary cover of the Lepontine dome (von
Eynatten, 2003). This was strongly supported by the very young zircon fission-track ages that match
the exhumation pattern of the Lepontine dome. However, garnet compositions in the youngest
Molasse sandstones are not comparable to Lepontine garnets sampled in this study nor to any detrital
garnet found in the main rivers draining the Lepontine dome today (Andò et al., 2014).
Instead, the occurrence of grossular- and spessartine-rich garnets in the 14 My-old Molasse mark a
distinct provenance change compared to the 19 My-old deposits that was not noticed in previous
studies (Schlunegger et al., 1998; Spiegel et al., 2000; von Eynatten, 2003). Garnets of this particular
composition are described from the Permo-Carboniferous plutons intruded into the crystalline
basement of the Aar and Mont Blanc massifs and the Gotthard nappe (Steck and Burri, 1971). Such
particular chemical composition provides a unique sedimentary fingerprint (Stutenbecker et al., 2017).
Their occurrence in the youngest Molasse sediments has important implications for the tectonic
evolution of the orogen. Until now, the surficial exposure of the external massifs in the Central Alps
was thought to post-date Molasse deposition. This interpretation relies principally on the absence of
pebbles of external massif origin (e.g. Aare granite) in the foreland basin (Trümpy, 1980). However,
many alpine granites closely resemble each other, especially if present as altered pebbles in the
Molasse deposits, and hence it is difficult to discount a specific source only on this basis. Further
support of late surficial exposure of the external massifs comes from structural reconstructions (e.g.
Pfiffner, 1986; 2017), that have located the top of the crystalline basement similar to the modern
topography, based on a relatively flat-lying contact between the crystalline basement and the overlying
Mesozoic sedimentary cover (Fig. 7a). According to this model and the published exhumation rates of
0.5-0.7 km/My (Michalski and Soom, 1990; Glotzbach et al., 2010), the top of the basement must have
been buried 7-10 km below the surface 14 Ma ago. However, Nibourel et al. (2018) have recently
proposed a revised geometry of the contact between crystalline basement and overlying cover, which
allows ca. 8 km of additional crystalline basement on top of the present-day topography (Fig. 7b). The
presence of external massif-sourced garnets in the youngest Molasse deposits provides independent
evidence that parts of the crystalline crust comprised in the external massifs were already at the



surface at ca. 14 Ma. Assuming the aforementioned average exhumation rates, 7-10 km of crystalline
basement would have already been exhumed (and subsequently eroded) during the past 14 My, which
is in good agreement with the geometric reconstructions by Nibourel et al. (2018).
5. Conclusions
Garnet geochemistry is a useful tool to further constrain the provenance of sediments in orogens such
as the Central Alps. We have demonstrated that it is possible to distinguish detrital garnets using a
combination of garnet classification schemes (Mange and Morton, 2007; Tolosana-Delgado et al.,
2018) and case-specific comparison with available alpine source rock compositions (Stutenbecker et
al., 2017). For the Miocene deposits of the Swiss Molasse basin we were able to (1) confirm the
provenance shift possibly related to the exhumation of the Lepontine dome between 25 and 19 My ago
as suggested by previous studies (von Eynatten, 2003) and (2) to identify an additional provenance
shift between ca. 19 and 14 My ago that had not been noticed before. The latter shift before 14 My ago
is related to the erosion of granites from the external crystalline massifs, which provides a minimum
age for their surficial exposure and corroborates their recently revised structural geometry (Fig. 7b).
Author contribution
LS designed the project. AM helped during field work and sample collection. PT and PL gave advice
for sample preparation, supported the microprobe measurements and data acquisition at the University
of Bern. LS prepared the manuscript with contributions by all co-authors.
Acknowledgements
This project was financially supported by a post-doctoral research grant awarded to L. Stutenbecker by
the International Association of Sedimentologists (IAS). We would like to thank Fritz Schlunegger for
guidance in the field and Alfons Berger and Lukas Nibourel for stimulating discussions.

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





*Table 1: Sample locations and characteristics of the Molasse sandstones from the Napf fan*

| Sample name | Sampling location | Lithostratigraphy (Matter, 1964; Schlunegger et al. 1996) | Magnetostratigraphic section (Schlunegger et al. 1996) | Magnetostratigraphic age (Schlunegger et al. 1996) |
|---|---|---|---|---|
| LS2017-3 | 47.00566 7.971325 | UFM, Napf beds | Fontannen section | ca. 14 Ma |
| LS2018-5 | 46.93913 7.950800 | UMM, Luzern formation | Schwändigraben section | ca. 19 Ma |
| LS2016-18 | 46.77463 7.732383 | LFM, Thun formation | Prässerebach section | ca. 25 Ma |

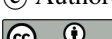



*Table 2: Sample locations and characteristics of potential source rocks*

| Sample name | Sampling location | River catchment | Metamorphic grade | Lithological unit |
|---|---|---|---|---|
| LS2018-12 | 46.72026 7.24548 | Ärgera, ca. 30 km$^2$ | Not metamorphic | Gurnigel flysch (detrital garnets) |
| LS2018-40 | 46.39026 8.54124 | Valle di Foioi, ca. 3 km$^2$ | Alpine amphibolite-facies | Orthogneiss of the Antigorio nappe, Lepontine dome |
| LS2016-43 | 46.43955 8.50115 | Valletta di Fiorina, ca. 8 km$^2$ | Alpine amphibolite-facies | Paragneiss of the Lebendun nappe, Lepontine dome |



*Table 3: Minimum, maximum and average contents (including standard deviation in brackets) of the*
*five common garnet endmembers in the Molasse sediments. For the full dataset we refer to*
*Stutenbecker (2019).*

| Sample | Minimum, maximum and average contents | Almandine (%) | Andradite (%) | Grossular (%) | Pyrope (%) | Spessartine (%) |
|---|---|---|---|---|---|---|
| **25 My** n=110 | min | 5 | 0 | 0 | 0 | 1 |
| | max | 86 | 48 | 40 | 22 | 38 |
| | average (standard deviation) | 70 (12) | 2 (5) | 9 (7) | 9 (5) | 9 (8) |
| **19 My** n=88 | min | 0 | 0 | 1 | 0 | 0 |
| | max | 84 | 91 | 55 | 39 | 39 |
| | average (standard deviation) | 65 (16) | 3 (13) | 16 (12) | 9 (8) | 5 (6) |
| **14 My** n=77 | min | 24 | 0 | 5 | 0 | 1 |
| | max | 74 | 9 | 60 | 26 | 39 |
| | average (standard deviation) | 50 (12) | 2 (2) | 32 (11) | 6 (5) | 9 (9) |




*Table 4: Results from classification following Mange & Morton (2007) and Tolosana-Delgado et al.*
*(2018). Using the linear discriminant method of Tolosana-Delgado et al. (2018) garnets were*
*attributed to one single class if the probability for that class was ≥50 %. Several grains were assigned*
*mixed probabilities with <50 % per class; these are listed separately below.*

| | Mange & Morton (2007) | | | | Tolosana-Delgado et al. (2018) | | |
|---|---|---|---|---|---|---|---|
| Types after Mange & Morton (2007) | **25 My** | **19 My** | **14 My** | Classes after Tolosana-Delgado et al. (2018) | **25 My** | **19 My** | **14 My** |
| Ci-type (high-grade metabasic) | | 5 % | 15 % | Eclogites (Class A) | | 1 % | |
| B-type (amphibolite facies) | 96 % | 84 % | 80 % | Amphibolites (Class B) | 71 % | 81 % | 78 % |
| A-type (granulite-facies) | 3 % | 8 % | | Granulites (Class C) | | 9 % | 5.5 % |
| D-type (metasomatic) | 1 % | 3 % | 5 % | Igneous (Class E) | 5 % | 3 % | 12 % |
| | | | | *Mixed probabilities Classes B-C* | *1 %* | *1 %* | |
| | | | | *Mixed probabilities Classes A-B-C* | | *5 %* | *4.5 %* |

636
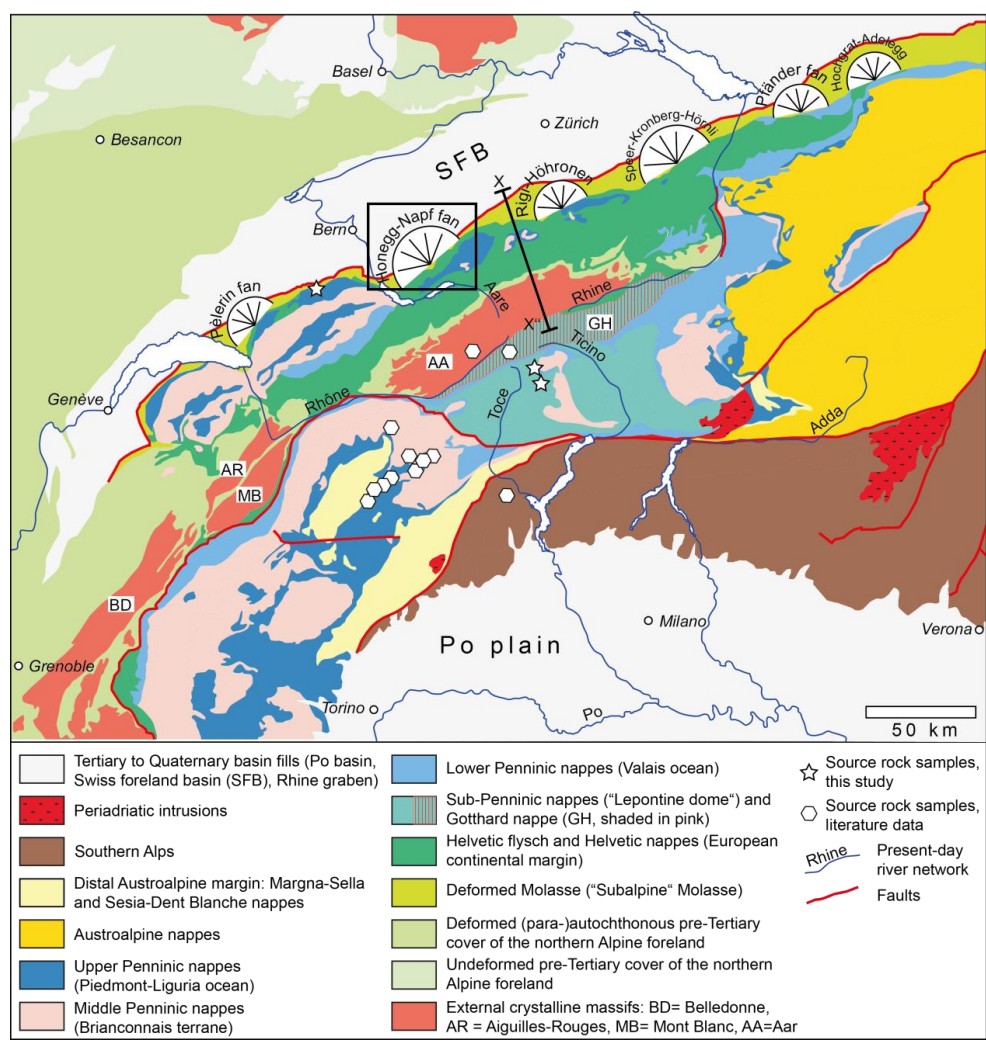

Fig. 1: *Simplified tectonic map of the Central Alps after Schmid et al. (2004) highlighting the location of alluvial fan deposits within the northern alpine foreland basin as well as the most important source rock units in the hinterland. The Napf fan, marked by the black rectangle, is located in the central part of the Swiss foreland basin (SFB).*



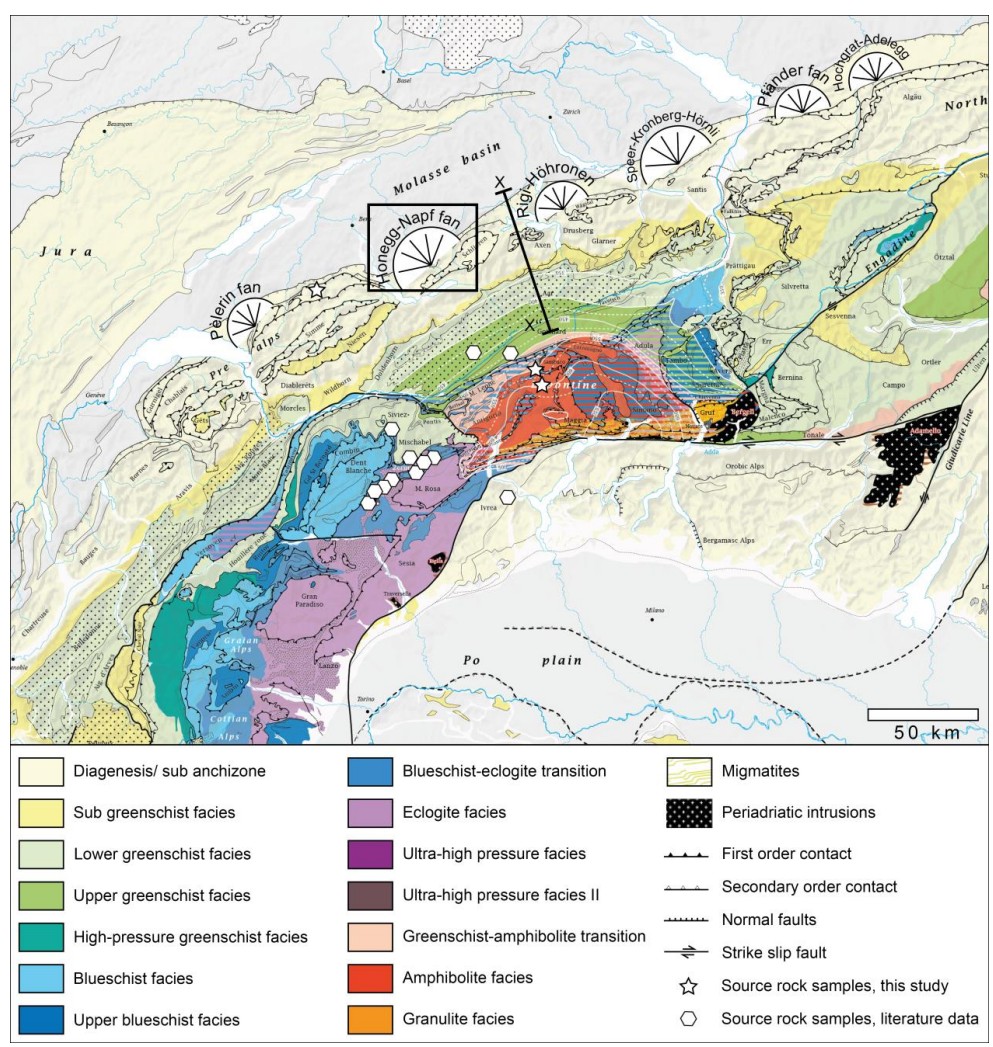

*Fig. 2: Metamorphic map of the Central Alps (Bousquet et al., 2012) showing the distribution and grade of alpine metamorphism. Note the increase from north to south from lower greenschist- to eclogite-facies conditions.*



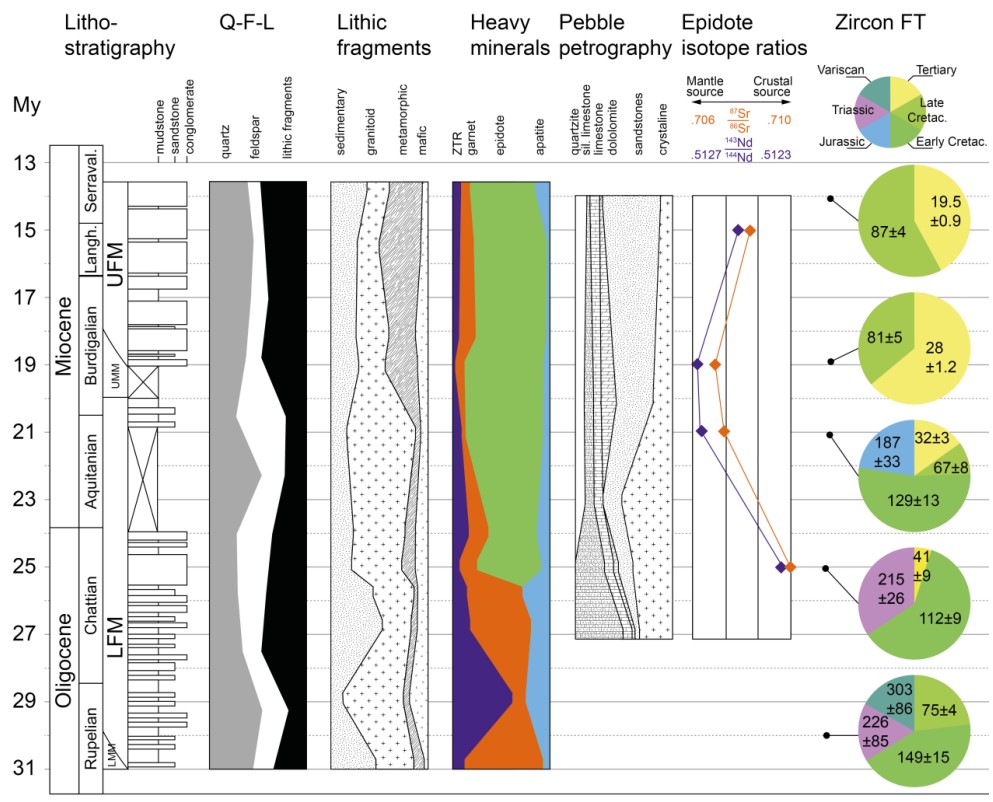

Fig. 3: Compilation of published compositional data in the Honegg-Napf fan. Heavy mineral and rock fragment data from the sand grain size after von Eynatten (2003), pebble petrography after Schlunegger et al. (1998), epidote isotope ratios after Spiegel et al. (2002) and zircon fission-track (FT) data after Spiegel et al. (2000).





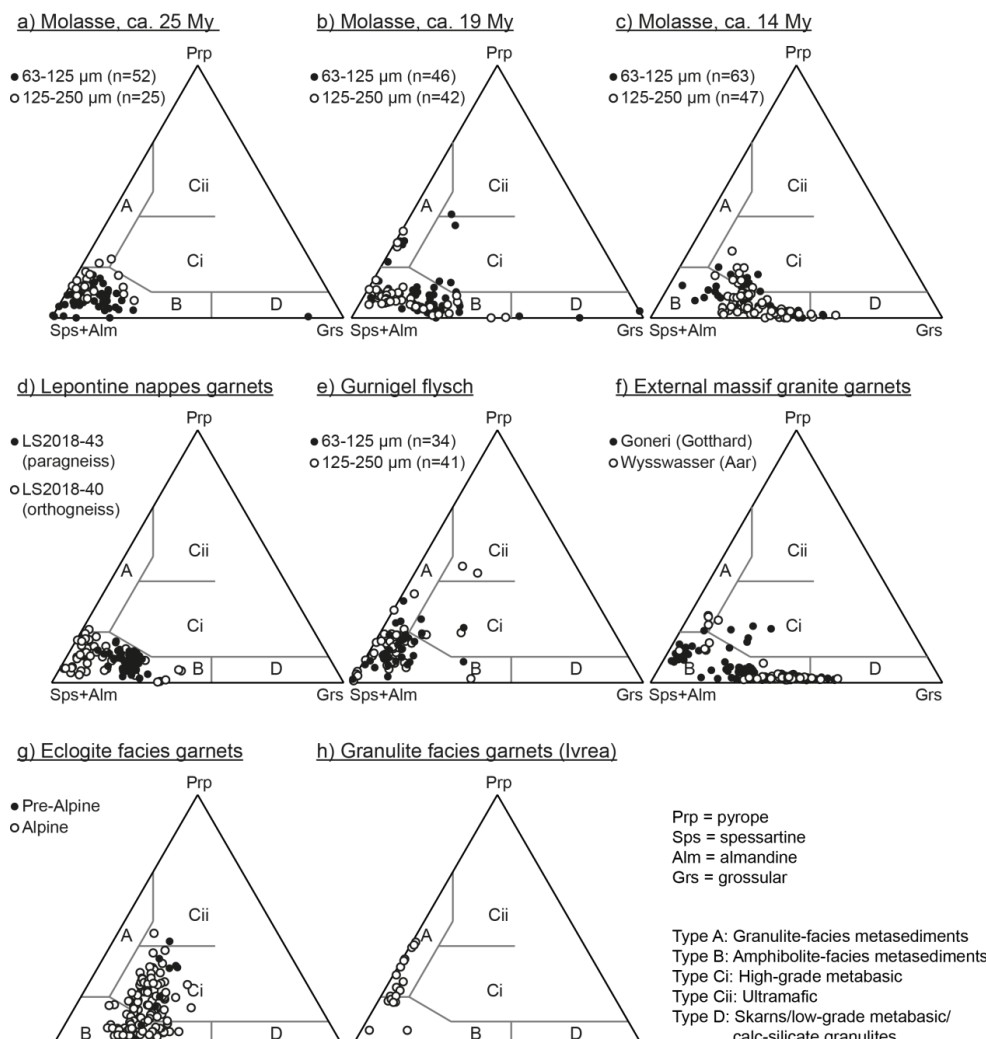

*Fig. 4: Ternary plots following the classification scheme of Mange & Morton (2007). (a-c) Garnet provenance changes in Molasse sandstones are marked by an increasing grossular content with decreasing age. Source rock data from (d) Lepontine gneisses (this study), (e) the Gurnigel flysch (this study), (f) external massif granitoids (Stutenbecker et al., 2017), (g) eclogite-facies rocks (Chinner & Dixon, 1973; Ernst & Dal Piaz, 1978; Oberhänsli, 1980; Sartori, 1990; Thélin et al., 1990; Reinecke, 1998; Cartwright & Barnicoat, 2002; Angiboust et al., 2009; Bucher & Grapes, 2009; Weber & Bucher, 2015), (h) granulite-facies rocks from the Ivrea zone in the Southern Alps (Hunziker & Zingg, 1980).*





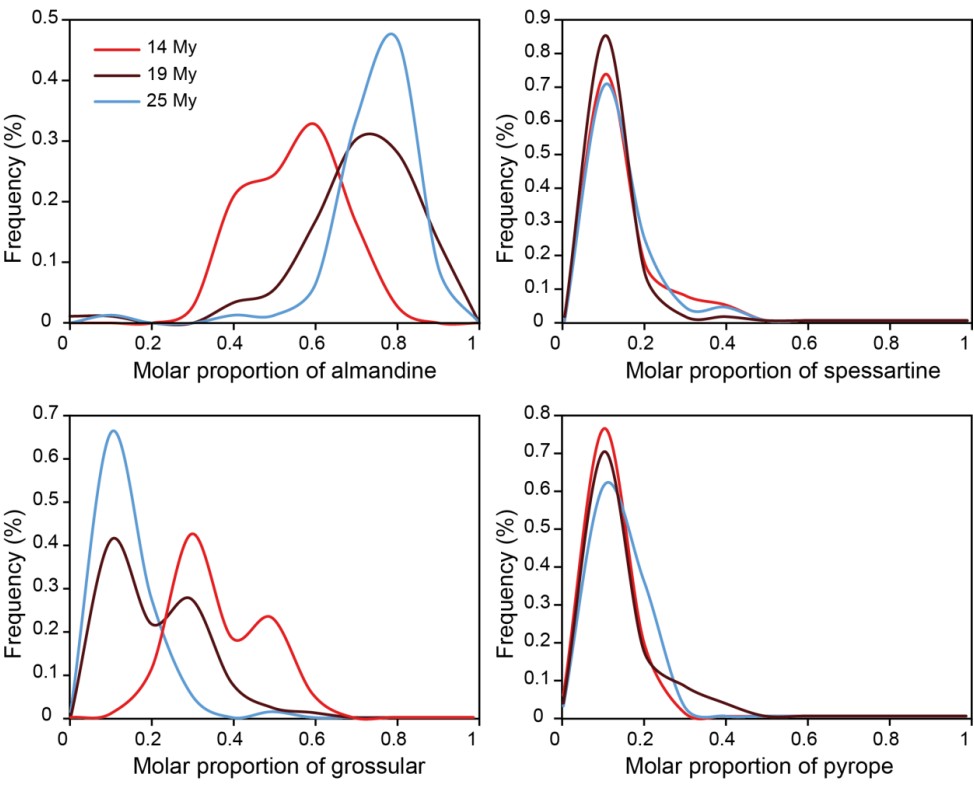

*Fig. 5: Shift of garnet compositions between the 25 My-, 19 My- and 14 My-old Molasse samples,*
*plotted as relative frequency of the four most common endmembers almandine, grossular, spessartine*
*and pyrope. While spessartine and pyrope contents are similar among the three samples, the*
*proportion of almandine decreases and the proportion of grossular increases with decreasing age.*
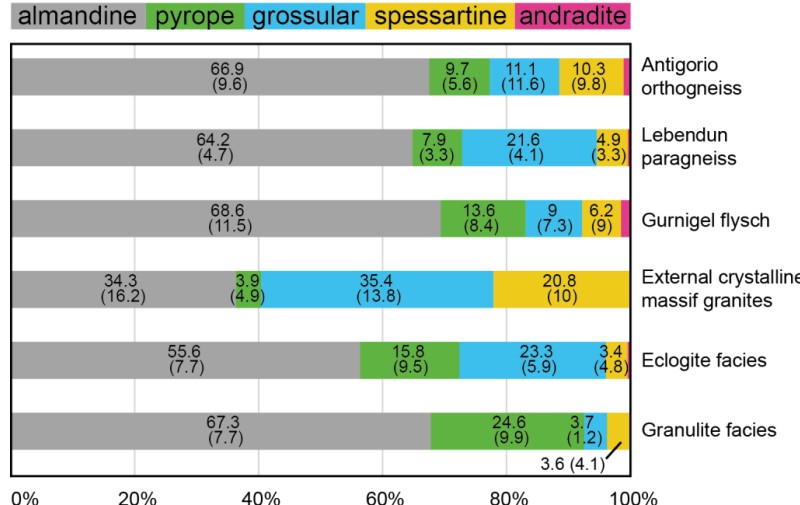

*Fig. 6: Average garnet compositions of potential source rocks from the Lepontine nappes (Antigorio orthogneiss and Lebendun paragneiss; this study), the Gurnigel flysch (this study), the external crystalline massif granites (Stutenbecker et al., 2017), eclogite facies rocks (Chinner & Dixon, 1973; Ernst & Dal Piaz, 1978; Oberhänsli, 1980; Sartori, 1990; Thélin et al., 1990; Reinecke, 1998; Cartwright & Barnicoat, 2002; Angiboust et al., 2009; Bucher & Grapes, 2009; Weber & Bucher, 2015), and granulite facies rocks (Hunziker & Zingg 1980). The means and standard deviations (in brackets) of each component are given for each bar.*



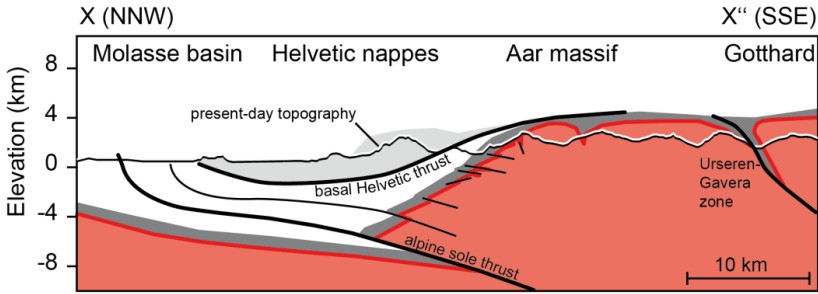

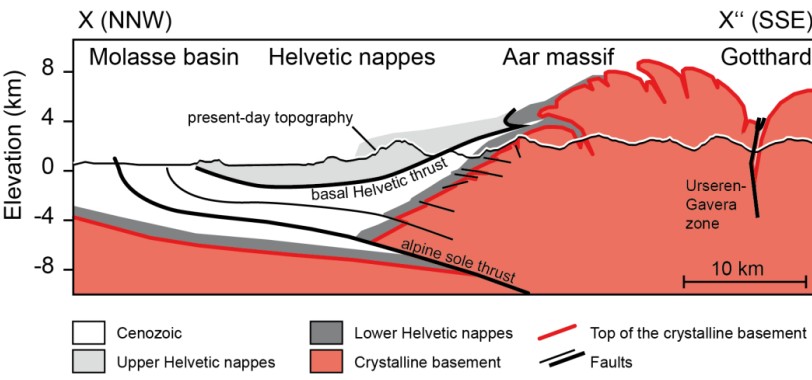

675

Fig. 7: Cross sections from X to X'' through the Aar massif simplified after Pfiffner (2017) and Nibourel et al. (2018). For trace of cross section see Fig. 1. (a): The reconstructed top of the crystalline basement in the Aar massif is located ca. 1-2 km higher than the present-day topography according to Pfiffner (2017). (b): In a revised version by Nibourel et al. (2018) the contact between the basement and the overlying Helvetic cover nappes is reconstructed to be steeper, resulting in ca. 8 km of (now eroded) crystalline crust on top of the present-day topography.