# Peer review of "Miocene basement exhumation in the Central Alps recorded"

_Solid Earth, 2019_

## Referee Comment (RC1) · Carita Augustsson (Referee) · 5 Jul 2019

The manuscript by Stutenbecker et al. with the title "Miocene basement exhumation in the Central Alps recorded by detrital garnet geochemistry in foreland basin deposits" is an Alpine provenance study based on garnet. The authors use the produced data to reveal source areas that have not been considered before, with implications for the exhumation history in the area. I have no scientific objections to the methodology, results quality or interpretations. My comments rather consider the text structure and how to improve the figures. Therefore, I estimate that only minor modifications are needed before publication in Solid Earth can be considered.

[Figure]

Below are my main comments. More detailed comments are in the manuscript file itself. I just realised that I somehow have managed to delete all my comments (not text modifications) up til Figure 3, so I have tried to reconstruct the most important ones. . .

1. Direct references to figures in the text The authors refer directly to figures in the normal text flow. This may cause a break in the reading. Therefore, I recommend rephrasing the text such that it rather focusses on the chemical composition of the analysed garnet than on discrimination fields in specific plots, see my example directly in the text.

2. Results in the Discussion chapter Some results are repeated in the Discussion chapter. This is unnecessary. Here, only the data should be interpreted, not presented again. I have marked such result entries in the Discussion chapter with green directly in the manuscript.

All other comments (n = 30) are written directly in the .doc manuscript file. This includes comments on the figures and the table. If you cannot read the file, please contact me (carita.augustsson@uis.no).

Although I am not a native English speaker, I have made some linguistic suggestions. I apologise if I have introduced any grammatical errors.

Please also note the supplement to this comment:
https://www.solid-earth-discuss.net/se-2019-98/se-2019-98-RC1-supplement.zip

---

## Referee Comment (RC2) · Lorenzo Gemignani (Referee) · 11 Jul 2019

Reviewer 2 comments

General reviewer comments: In the Paper titled "Miocene basement exhumation in the central Alps recorded by detrital garnet geochemistry in foreland basin deposits" Stutenbecker et al. use a relatively new provenance tool to infer a minimum peak age of the exhumation of the External Alpine Massifs and their consequent exposure as a surface lithologies. Their major outcomes highlight the possibility that portions of the external massifs have been exhumed and eroded since ~14 Ma. This could be regarded as a potential novel find and I think that is a good starting point to speculate on

the models of exhumation of the External Basement Massifs in the Alps. However, in my opinion, their work has a few new data to convince the audience that the onset of External Massifs Rocks has been driven during the mid-Miocene by high denudation coupled with crustal delamination and buoyancy-driven vertical uplift. They use this model as a key to interpreting their detrital data. This is, due to the lack of data is a bit redundantly stressed and needs to be reformulated. I, therefore, suggest the authors reworking the structure of their paper focussing in describing the previously proposed model with more objectivity with respect to their new data. I have tried to highlight two major points of weakness of this manuscript which I think the author might want to improve: First the paucity of new data, the authors present results from only three samples (and additional previously published data) comparing the chemistry of the garnet with the source rocks information (3 additional samples). This is a good pilot approach but needs more constraints, possibly expanding the area of investigation to different fan deposits in the foreland to gain confidence in drawing interpretation for the onset of exhumation and erosion of the External Massifs Units. Furthermore, I find that the authors lack while interpreting/presenting their detrital datasets of a correct acknowledgment and discussion of works that focussed on the present-day evolution of detrital thermochron/petrographic proxies in the Alps. I think that would be useful to compare other proxies available in the literature with garnet chemical composition. What other analytical detrital/in-situ methods describe? Second, the authors seem supporting "a priori" the model of "buoyancy-driven vertical displacement" associated with slab dynamics and erosional unloading, as a prerequisite to interpret their dataset (e.g. Herwegh et al., 2017; Nibourel et al., 2018). Those models and other proposed interpretations could, in my opinion, be described in more detail in the introduction, whereas in the discussion the authors reconcile their data with the geometric interpretation of Nibourel et al. (2018). This is an interesting ongoing discussion and might be expanded (e.g. Herman et al., 2013, Herwegh et al., 2017, Schildgen et al., 2018). I would suggest redrawing your discussion by inserting yours and available literature data in a more precise metamorphic, tectonic and erosional patterns context. The latest, in my opinion, would require a bit of discussion on how the foreland deposits might have been biased by e.g. river patterns reorganization during Miocene to present-day time, heterogeneous erosional patterns along strike, glacial processes, etc. Those processes are important for the evolution of the detrital record and need to be accounted while interpreting provenance data. It would be really helpful to show a compilation of different available datasets as a map view tracking External Massifs source units and their contribution in the Molasse sedimentary deposits. How does the hinterland info's are correlated with the detrital ones? A Map would greatly help the reader to track source hinterland and detrital provenance, the author cuould benefit by using their previous work e.g. Stutenbecker et al. (2017). An effort has been done in Figure. 2. However, there is not a correspondence between the legend and metamorphic grade indicated in the map. This map might be redrawn as a simplified map highlighting the information that is essential to understand the authors' discussion. Overall, the paper reads well but there are a few changes required. I have noticed a few interferences between results description and discussion, this might be changed. The English language is good, although I might not be the best example of scrutinizer on this topic, I, therefore, suggest a native English colleague reading the manuscript once.

Comments byline: 25. "Tectonic processes influence" I find "influence" a bit week, maybe change with "regulate" or "drive" the evolution of mountain chains. 34. Please be more specific, what you mean for highest erosion rates in the Alps in (mm/yr) or as you mention in line 43 km/Myr. 61. New provenance studies that used detrital thermochronology multi-proxy approach to constrain exhumation rates and its spatial variability has been recently used in the Alps (e.g. Carrapa et al., 2016; Tectonics; Gemignani et al., 2017. Tectonics) and need to be acknowledged. 72-75. Additional information to what. Does the author mean to previously published papers? Such as for instance Stutenbecker et al. (2017). Tectonic forcing of the Molasse basin or in the hinterland? Please be more specific. 82-84. Reference is needed 105. architectural elements are capital, column, architrave, etc. Do the authors mean tectonic units or litho-tectonic units? 119-120: It would be useful if the author could refer to a temporal

frame when invoking for timing and rates comparing it with other's colleague works. This will help the reader to follow the argumentation in chronologic order. 106-142. What is the relationship of this description of the potential source rocks with the garnet composition? This is important for a clear understanding of the relationship between hinterland source units and syn-sedimentary sequences in the foreland. I think would be worth to expand this description with a map or figure showing potential source in the hinterland and their present-day distribution in the foreland units. 143. The Napf fan It is the first time that this fan is mentioned in the text. This information is missing in section 1 and should be introduced before in the text. 208. Fertility is a specific definition applied to detrital sediments. Please make sure you properly introduce this concept and acknowledge the promotors of this new definition. 213. What is the effect that you might obtain by using pestle and mortar on the round-shaped grains of garnets? There is not a less invasive mineral separation technique? 228-229. This might be related to an incorrect mineral separation approach and mislead to biased interpretation of the data. How could you check for consistency of the data? In other words, how fractures might bias your chemical analysis? Please explain. 229. Could the authors specify the amount of "randomly selected grains"? 246. figure 4 is confusing because the authors use black and white tones to indicate a different aspect of the different ternary plots. This could be improved by using a colored version of the figure with a color-coded legend. 272-275. Here, you are discussing the data. Please objectively describe the data. 295-297. Here, you are presenting results. Please reformulate this sentence. 348-354. The authors describe their data but what is lacking, in my opinion, is a clear discussion of what is the importance of those data for interpreting the evolution of the External Basement Massifs. In particular, I think that would be really interesting to insert this new preliminary finding i.d. the External Massifs Units reached the surface at ~14 Ma as constrained by Grn chemical composition, in relation with the thermokinematic model of low-temperature chronometers arguing for a sustained increase of denudation during the Pliocene. This has been the focus of a recent debate in literature see e.g. Schildgen et al. 2018 vs. Fox et al. 2015,

2016, Herman et al., 2013, etc., and I think it is important to discuss it. 363-364. What is the present-day evolution of the detrital provenance/thermochronological signal? Which units constitute the present-day major erosional contributions in the Alpine river patterns? I think that might be useful for the authors to acknowledge recent studies that worked on tracking source rocks information with detrital thermochronologic evolution of modern river sands in the Alpine river patterns. There are several works that investigate these processes in a different portion of the Alps and should be, in my opinion, acknowledged (Bernet et al., 2009, Carrapa et al., 2004, Gemignani et al., 2017; Resentini et al., 2012). 365. "Very young", how young <2 Ma, <5 Ma, <10 Ma, <30 Ma? 370-393. At this point, it is clear that the compositional change of the garnets in the youngest ∼14 Ma foreland deposits with respect to the older ∼19 Ma interval (where Grn yield a different composition = different provenance) has been interpreted by the authors as the lower temporal limit for the surficial exposure of the External Basement Massifs units. Using this new observation they argue for "important implication for the tectonic evolution of the orogen" (Lines 375-376). Furthermore, the authors support the geometric restoration of the central Alps (Aar Massif-Helvetic nappes) as proposed by Nibourel et al., 2018, where ∼7-8 km of basement rocks have been exhumed and eroded since ∼14 Ma lead by "lithospheric mantle roll back" associated with "crustal delamination" and "buoyancy-driven vertical exhumation coupled with surface erosion" of the External Basement Massifs (e.g. Herwegh et al., 2017). This point in the discussion is clear and well expressed, however, I think that you should describe also the other proposed model in the introduction, and, lately, data on hands, describe why your data support this proposed hypothesis. This is, in my opinion, a bit lacking in the text and would require some improvements.

Please also note the supplement to this comment:
https://www.solid-earth-discuss.net/se-2019-98/se-2019-98-RC2-supplement.pdf

---

## Author Comment (AC1) · 14 Aug 2019

We thank Carita Augustsson for her thorough review.

We have prepared three files to answer to her comments: 1) Reply to her general comments (a pdf file called "Response to reviewers-Augustsson") 2) Reply to her detailed comments (directly in the doc file she prepared- called "Answer to Comments") 3) A revised version of the manuscript showing the changes we made based on suggestions of both reviewers, including updated tables and figures.

Please find all three files in the attached zip file.

[Figure]

Please also note the supplement to this comment:
https://www.solid-earth-discuss.net/se-2019-98/se-2019-98-AC1-supplement.zip
* * *

---

## Author Comment (AC2) · 14 Aug 2019

We thank Lorenzo Gemignani for his constructive review.

We have prepared two files to answer to his comments: 1) Reply to his comments (a pdf file called "Response to reviewers-Gemignani") 2) A revised version of the manuscript showing the changes we made based on suggestions of both reviewers, including updated tables and figures.

Please find both files in the attached zip file.

[Figure]

Please also note the supplement to this comment:
https://www.solid-earth-discuss.net/se-2019-98/se-2019-98-AC2-supplement.zip

---

## Author Response (AR1)

**Point-by-point response to reviews (replies marked in blue)**

Referee #1: Carita Augustsson

The manuscript by Stutenbecker et al. with the title "Miocene basement exhumation in the Central Alps recorded by detrital garnet geochemistry in foreland basin deposits" is an Alpine provenance study based on garnet. The authors use the produced data to reveal source areas that have not been considered before, with implications for the exhumation history in the area. I have no scientific objections to the methodology, results quality or interpretations. My comments rather consider the text structure and how to improve the figures. Therefore, I estimate that only minor modifications are needed before publication in Solid Earth can be considered.

Reply: Thank you for the thorough review. We have addressed all changes to the figures as requested. Several suggestions concerned the discussion section, because it contained some repetitions from the results section. Accordingly, we restructured the discussion and eliminated the repetitions (see further below).

Below are my main comments. More detailed comments are in the manuscript file itself. I just realised that I somehow have managed to delete all my comments (not text modifications) up til Figure 3, so I have tried to reconstruct the most important ones. . .

    1. Direct references to figures in the text

The authors refer directly to figures in the normal text flow. This may cause a break in the reading. Therefore, I recommend rephrasing the text such that it rather focusses on the chemical composition of the analysed garnet than on discrimination fields in specific plots, see my example directly in the text.

Reply: We have removed these direct references as suggested.

    2. Results in the Discussion chapter

Some results are repeated in the Discussion chapter. This is unnecessary. Here, only the data should be interpreted, not presented again. I have marked such result entries in the Discussion chapter with green directly in the manuscript.

Reply: We agree. We have restructured the discussion extensively based on suggestions by both reviewers. Instead of using the old structure "origin of amphibolite-facies garnets", "origin of eclogite-facies garnets" etc. we have created the new paragraphs "Late Oligocene (~25 My)", "Early Miocene (~19 My)" and "Middle Miocene (~14 My)". Like this, we focus the important information in a better way and avoid the repetition and the interference of results and discussion. This new discussion is structured around the new Fig. 6, which is a summary of the paleogeography of the Central Alpine hinterland and the catchment of the Napf fan.

All other comments (n = 30) are written directly in the .doc manuscript file. This includes comments on the figures and the table. If you cannot read the file, please contact me (carita.augustsson@uis.no).

Although I am not a native English speaker, I have made some linguistic suggestions. I apologise if I have introduced any grammatical errors.

Reply: All spelling mistakes, linguistic improvements and "minor" corrections (e.g. the use of "sedimentary rock" instead of "sediment", "Alpine" instead of "alpine", the use of the singular mineral/rock instead of the plural, etc.) were accepted using the track changes function directly in the revised document. Please see the revised version of the manuscript for all implemented changes (including the ones from reviewer #2 and our own corrections). For the answers to the detailed comments/suggestions, see the additional supplementary file ("Answer to Comments").

General reviewer comments:

In the Paper titled "Miocene basement exhumation in the central Alps recorded by detrital garnet geochemistry in foreland basin deposits" Stutenbecker et al. use a relatively new provenance tool to infer a minimum peak age of the exhumation of the External Alpine Massifs and their consequent exposure as a surface lithologies. Their major outcomes highlight the possibility that portions of the external massifs have been exhumed and eroded since ~14 Ma. This could be regarded as a potential novel find and I think that is a good starting point to speculate on the models of exhumation of the External Basement Massifs in the Alps. However, in my opinion, their work has a few new data to convince the audience that the onset of External Massifs Rocks has been driven during the mid-Miocene by high denudation coupled with crustal delamination and buoyancy-driven vertical uplift. They use this model as a key to interpreting their detrital data. This is, due to the lack of data is a bit redundantly stressed and needs to be reformulated. I, therefore, suggest the authors reworking the structure of their paper focussing in describing the previously proposed model with more objectivity with respect to their new data.
Reply: Thank you for your review. Please see the reply to the comments below.

I have tried to highlight two major points of weakness of this manuscript which I think the author might want to improve:

First the paucity of new data, the authors present results from only three samples (and additional previously published data) comparing the chemistry of the garnet with the source rocks information (3 additional samples). This is a good pilot approach but needs more constraints, possibly expanding the area of investigation to different fan deposits in the foreland to gain confidence in drawing interpretation for the onset of exhumation and erosion of the External Massifs Units. Furthermore, I find that the authors lack while interpreting/presenting their detrital datasets of a correct acknowledgment and discussion of works that focussed on the present-day evolution of detrital thermochron/petrographic proxies in the Alps. I think that would be useful to compare other proxies available in the literature with garnet chemical composition. What other analytical detrital/in-situ methods describe?
Reply: We are aware that further data from additional samples could affirm our interpretation. However, the presence of the described "exotic" garnet, even if only in one sample, proves that a source supplying those garnets must have been exposed in the hinterland. We thoroughly reviewed other provenance proxies from the study area (von Eynatten, 2003; Spiegel et al., 2000; 2003, amongst others, see Fig. 3). The problem with other proxies is that they are not unambiguous and could be interpreted in different ways, as explained in the text. I understand that the reviewer has a thermochronology background and therefore misses a more detailed discussion of the thermochronological data available. The problem with thermochronology in this setting is that we do not have bedrock data available to compare the detrital ZFT distributions (Spiegel et al. 2002) to. Because the top of the external massif has been eroded, bedrock thermochronological data are obviously not available anymore, and the oldest grains in the external massifs are around 21My (ZFT) and 10My (AFT) old. We cannot know what happened before and what kind of FT ages the (now eroded) part of the external massifs would have supplied and what that would imply in terms of exhumation rates. In contrast, garnet that is only found in specific lithologies of specific metamorphic grade provides direct evidence and its presence is largely independent from assumptions on factors such as geothermal gradients and closing temperatures. We do not see the benefit of including modern-day thermochronological data in this context.

Second, the authors seem supporting "a priori" the model of "buoyancy-driven vertical displacement" associated with slab dynamics and erosional unloading, as a prerequisite to interpret their dataset (e.g. Herwegh et al., 2017; Nibourel et al., 2018). Those models and other proposed interpretations could, in my opinion, be described in more detail in the introduction, whereas in the discussion the authors reconcile their data with the geometric interpretation of Nibourel et al. (2018). This is an interesting ongoing discussion and might be expanded (e.g. Herman et al., 2013, Herwegh et al., 2017, Schildgen et al., 2018).

Reply: As suggested we have included a brief description of alternative models into the introduction (lines 38-42 as well as 53-57 in the revised version). The discussion on climatic effects and glacial erosion led by authors such as Herman and Schildgen concerns the late Neogene (essentially <5 My). Whether or not the late Neogene cooling had an effect on exhumation rates (and consequently erosion and sediment accumulation) is interesting, but is not directly linked to our study, because Molasse deposition ceases around 14 My ago. We do not claim that the Herwegh model is the only explanation for exhumation and erosion in the Alps and we do not exclude that climatic changes are important as well. Our study improves our understanding of the timing (onset) of exhumation rather than the process that is causing it.

I would suggest redrawing your discussion by inserting yours and available literature data in a more precise metamorphic, tectonic and erosional patterns context. The latest, in my opinion, would require a bit of discussion on how the foreland deposits might have been biased by e.g. river patterns reorganization during Miocene to present-day time, heterogeneous erosional patterns along strike, glacial processes, etc. Those processes are important for the evolution of the detrital record and need to be accounted while interpreting provenance data.

Reply: The aim of this study was to test whether or not detritus from the external massifs is present in the Molasse and whether or not this tectonic unit should be considered as a sediment source already in the Miocene. The aim of this study was not the review of the paleogeography or the drainage development during the Miocene. This would, as the reviewer points out, need further data from other locations and also other provenance proxies. We have, however, included a figure showing the paleogeography of the study area at the different time slices that show the interpreted drainage divide (new Fig. 6). We have essentially used published models for this, and have applied changes according to our new findings.

Glacial processes are not relevant in the Oligo- and Miocene.

It would be really helpful to show a compilation of different available datasets as a map view tracking External Massifs source units and their contribution in the Molasse sedimentary deposits. How does the hinterland info's are correlated with the detrital ones? A Map would greatly help the reader to track source hinterland and detrital provenance, the author cuould benefit by using their previous work e.g. Stutenbecker et al. (2017). An effort has been done in Figure. 2. However, there is not a correspondence between the legend and metamorphic grade indicated in the map. This map might be redrawn as a simplified map highlighting the information that is essential to understand the authors' discussion.

Reply: As suggested we simplified Fig. 1 and 2 to make it easier for the reader to follow the discussion. We added some important names to Fig. 1 (Lepontine dome, Prealps Romandes,

…) to guide the reader. We also added the fission track data from Bernet et al. (2009) into a new map (Figure 2b)

If I understood the reviewer's comment correctly, he asks for a map or several maps showing the evolution of the hinterland through time, so basically a paleogeographic reconstruction. We have added an interpretation of the paleogeography (new figure 6). This figure is based on previous interpretations and we have added some changes according to our new findings.

Overall, the paper reads well but there are a few changes required. I have noticed a few interferences between results description and discussion, this might be changed. The English language is good, although I might not be the best example of scrutinizer on this topic, I, therefore, suggest a native English colleague reading the manuscript once.

Reply: We have revised the discussion and restructured it to eliminate the described interferences between results and discussion. Co-author Peter Tollan is a native speaker and he has carefully reviewed this manuscript.

Comments by line:

25. "Tectonic processes influence" I find "influence" a bit week, maybe change with "regulate" or "drive" the evolution of mountain chains.

Reply: We replaced "influence" by "drive"

34. Please be more specific, what you mean for highest erosion rates in the Alps in (mm/yr) or as you mention in line 43 km/Myr.

Reply: Erosion rates in the Aar massif are >1mm/y as presented in Wittmann et al. (2007) and Stutenbecker et al. (2018). We modified accordingly: "…the highest denudation rates measured in the Alps (up to 1.4 mm/y), which all contribute…" (line 36 in the revised version)

61. New provenance studies that used detrital thermochronology multi-proxy approach to constrain exhumation rates and its spatial variability has been recently used in the Alps (e.g. Carrapa et al., 2016; Tectonics; Gemignani et al., 2017. Tectonics) and need to be acknowledged.

Reply: These do not concern the Molasse deposits in the Central Alps (Carrapra et al. worked on the Western Alps, and Gemignani et al. studied modern rivers). Perhaps the reviewer could be more specific on the value of these data for our interpretation.

72-75. Additional information to what. Does the author mean to previously published papers? Such as for instance Stutenbecker et al. (2017). Tectonic forcing of the Molasse basin or in the hinterland? Please be more specific.

Reply: No, we do not mean Stutenbecker et al. (2017), which is not a Molasse-related study. We have rephrased this sentence: "We aim (1) to explore if detrital garnet geochemistry can help identifying additional provenance changes in the Miocene Molasse deposits that have gone unnoticed so far and (2) to test whether detritus from the external massifs is present in the younger Molasse deposits in order to give independent constraints on the timing of crystalline basement exhumation." (lines85-88 in the revised version)

82-84. Reference is needed

Reply: We added Allen et al., 1991 and Sinclair, 1997 (line 100 in the revised version)

105. architectural elements are capital, column, architrave, etc. Do the authors mean tectonic units or litho-tectonic units?

Reply: We replaced "architectural elements" by "tectonic units". Please note, however, that "architectural elements" is frequently used also in geological contexts (e.g. in sedimentology, Miall (1985)).

119-120: It would be useful if the author could refer to a temporal frame when invoking for timing and rates comparing it with other's colleague works. This will help the reader to follow the argumentation in chronologic order.

Reply: I do not understand this. The history of the burial and exhumation of the external massifs is reported here in chronologic order. We mention the exhumation rates already before (line 52 in the revised version).

106-142. What is the relationship of this description of the potential source rocks with the garnet composition? This is important for a clear understanding of the relationship between hinterland source units and syn-sedimentary sequences in the foreland. I think would be worth to expand this description with a map or figure showing potential source in the hinterland and their present-day distribution in the foreland units.

Reply: In this section we introduce the tectonic units to readers not familiar with the Alps. The most relevant information here for the later interpretation of the garnet chemistry is the metamorphic grade of the units, which the later interpretation relies on.
We simplified the maps in Fig. 1 and 2 to show only the primary tectonic units mentioned in this paragraph and eliminated/simplified some unnecessary details. However, I do not understand what the reviewer means by "their present-day distribution in the foreland units". The tectonic units are in the hinterland, not in the foreland… The distribution of what?

143. The Napf fan It is the first time that this fan is mentioned in the text. This information is missing in section 1 and should be introduced before in the text.

Reply: I am not sure I agree with this. In this paragraph we introduce the fan system properly and justify why we chose this one in particular. This is a subsection to section 1 and I do not see the need to introduce it before.

208. Fertility is a specific definition applied to detrital sediments. Please make sure you properly introduce this concept and acknowledge the promotors of this new definition.

Reply: This is false. Fertility is not used in sediments (this would be the heavy mineral concentration, see Garzanti & Andò, 2007), but it is a measure of the abundance of a specific mineral in a source rock (e.g. Malusà et al. 2016). We do not see the necessity of introducing this concept, as it is not relevant for the study. Instead, we refer to the review of Malusà et al., (2016) for more information.

213. What is the effect that you might obtain by using pestle and mortar on the round-shaped grains of garnets? There is not a less invasive mineral separation technique?

Reply: The sandstones we collected were not well cemented and disaggregated easily without applying force. We have not noticed much difference to other separation techniques. If any, the pestle + mortar technique produces less dust than a jaw breaker/ mill, indicating that it is destroying the particles less, at least in this kind of sandstone.

228-229. This might be related to an incorrect mineral separation approach and mislead to biased interpretation of the data. How could you check for consistency of the data? In other words, how fractures might bias your chemical analysis? Please explain.

Reply: Cracked grains were an exception and in >95% of grains we were able to measure the center of the grain. Fractures in garnets do not influence the chemical composition!

229. Could the authors specify the amount of "randomly selected grains"?

Reply: These were 22 core-and-rim-pairs. See supplementary material for details.

246. figure 4 is confusing because the authors use black and white tones to indicate a different aspect of the different ternary plots. This could be improved by using a colored version of the figure with a color-coded legend.

Reply: We modified the figure accordingly.

272-275. Here, you are discussing the data. Please objectively describe the data.

Reply: We removed this sentence.

295-297. Here, you are presenting results. Please reformulate this sentence.

Reply: We removed this sentence as we restructured the discussion section.

348-354. The authors describe their data but what is lacking, in my opinion, is a clear discussion of what is the importance of those data for interpreting the evolution of the External Basement Massifs. In particular, I think that would be really interesting to insert this new preliminary finding i.d. the External Massifs Units reached the surface at ~14 Ma as constrained by Grn chemical composition, in relation with the thermokinematic model of low-temperature chronometers arguing for a sustained increase of denudation during the Pliocene. This has been the focus of a recent debate in literature see e.g. Schildgen et al. 2018 vs. Fox et al. 2015, 2016, Herman et al., 2013, etc., and I think it is important to discuss it.

Reply: We have expanded the introduction and added the suggested references to the introduction (lines 38-42 in the revised version). However, the Pliocene exhumation history is not related to our study, which is why we do not mention this in the discussion.

363-364. What is the present-day evolution of the detrital provenance/thermochronological signal? Which units constitute the present-day major erosional contributions in the Alpine river patterns? I think that might be useful for the authors to acknowledge recent studies that worked on tracking source rocks information with detrital thermochronologic evolution of modern river sands in the Alpine river patterns. There are several works that investigate these processes in a different portion of the Alps and should be, in my opinion, acknowledged (Bernet et al., 2009, Carrapa et al., 2004, Gemignani et al., 2017; Resentini et al., 2012).

Reply: We have included the modern-day bedrock ages from Bernet et al. 2009 to Figure 2 (Figure 2b) and now refer to this publication when comparing the detrital fission track ages from Spiegel et al. (2000) to potential sources. All of the other suggested references deal with thermochronology in different areas of the orogen (Western Alps, retro-foreland, axial drainage of the northern foreland basin). I do not see how this would help our interpretation. Perhaps the reviewer could be more specific on the value of these data for our interpretation.

365. "Very young", how young <2Ma, <5 Ma, <10 Ma, <30 Ma?

Reply: According to Spiegel et al. (2000) the youngest age peak in Molasse of this age is 19.5±0.9 My. We added this information to lines 195 and 415 in the revised version.

370-393. At this point, it is clear that the compositional change of the garnets in the youngest ~14 Ma foreland deposits with respect to the older ~19 Ma interval (where Grn yield a different composition = different provenance) has been interpreted by the authors as the lower temporal limit for the surficial exposure of the External Basement Massifs units. Using this new observation they argue for "important implication for the tectonic evolution of the orogen" (Lines 375-376). Furthermore, the authors support the geometric restoration of the central Alps (Aar Massif-Helvetic nappes) as proposed by Nibourel et al., 2018, where ~7-8 km of basement rocks have been exhumed and eroded since ~14 Ma lead by "lithospheric mantle roll back" associated with "crustal delamination" and "buoyancy-driven vertical exhumation coupled with surface erosion" of the External Basement Massifs (e.g. Herwegh et al., 2017). This point in the discussion is clear and well expressed, however, I think that you should describe also the other proposed model in the introduction, and, lately, data on hands, describe why your data support this proposed hypothesis. This is, in my opinion, a bit lacking in the text and would require some improvements

Reply: As already pointed out in the second comment, we have added the alternative exhumation models in the introduction (lines 38-42 in the revised version). Our results do not improve our understanding of the process of exhumation, but that of its timing. As pointed out before, we cannot reconstruct the exhumation rates, because we lack bedrock data of what has already been eroded, and we can therefore not say if exhumation was faster or slower in the Miocene.

**List of all relevant changes made to the manuscript**

- Text:
  - Introduction: We have added some relevant references that concern the ongoing discussion on the exhumation mechanism in the study area (mantle processes vs. erosional unloading and related climatic triggers) as requested by by referee #2.
  - Discussion: We have reorganized the discussion section extensively. Both referees pointed out that we had some interference between the results and discussion sections and that some points were repeated. Before, the sub-sections were describing the origin of individual garnet groups ("origin of eclogite facies garnets", "origin of amphibolite facies garnets" etc.) and then there was a final sub-section showing the implications to the timing of exhumation. Now, we have reduced the subsections to three that are organized around the new Fig. 6 (showing the paleogeography, see below) and discuss the implications of our results to the respective time slices at 25 My, 19 My and 14 My ago.
- Tables:
  - We have extended Table 3 to show information that was previously contained in Fig. 6 (now deleted, see below)
- Figures:
  - We have simplified Fig. 1 by combining some tectonic units, adding black lines to their outlines and we added the names of some important tectonic units that we mention in the text
  - We have simplified Fig. 2 (now Fig. 2a) by removing some unnecessary details. We used an easier color scheme and added the outlines of tectonic units (in Fig. 1) to facilitate the comparison with Fig. 1.
  - We have added a new Fig (Fig. 2b) showing the zircon fission track ages of bedrock exposed in the Central Alps (as suggested by referee #2)
  - We have changed Fig. 4 from greyscales to a colored version (as suggested by referee #2)
  - We have deleted the former Fig. 6 and instead combined the information contained in that figure into Table 3 (as suggested by referee #1)
  - We have prepared a new Fig. 6 showing a reconstruction of the paleogeography and the source areas present at each time slice (as suggested by referee #2)

[revised manuscript text omitted]